# Continuous-Wave Pumped Monolayer WS_2_ Lasing for Photonic Barcoding

**DOI:** 10.3390/nano14070614

**Published:** 2024-03-30

**Authors:** Haodong Cheng, Junyu Qu, Wangqi Mao, Shula Chen, Hongxing Dong

**Affiliations:** 1Key Laboratory for Micro-Nano Physics and Technology of Hunan Province, College of Materials Science and Engineering, Hunan University, Changsha 410082, China; haodongc@hnu.edu.cn (H.C.); qujunyu@hnu.edu.cn (J.Q.); 2Hunan Institute of Optoelectronic Integration, Hunan University, Changsha 410082, China; 3Key Laboratory of Materials for High-Power Laser, Shanghai Institute of Optics and Fine Mechanics, Chinese Academy of Sciences, Shanghai 201800, China; 4Hangzhou Institute for Advanced Study, Chinese Academy of Sciences, Hangzhou 310024, China

**Keywords:** micro/nano photonic barcode, anti-counterfeiting, monolayer WS_2_ films, continuous-wave pumped lasing, high encoding capacity

## Abstract

Micro/nano photonic barcoding has emerged as a promising technology for information security and anti-counterfeiting applications owing to its high security and robust tamper resistance. However, the practical application of conventional micro/nano photonic barcodes is constrained by limitations in encoding capacity and identification verification (e.g., broad emission bandwidth and the expense of pulsed lasers). Herein, we propose high-capacity photonic barcode labels by leveraging continuous-wave (CW) pumped monolayer tungsten disulfide (WS_2_) lasing. Large-area, high-quality monolayer WS_2_ films were grown via a vapor deposition method and coupled with external cavities to construct optically pumped microlasers, thus achieving an excellent CW-pumped lasing with a narrow linewidth (~0.39 nm) and a low threshold (~400 W cm^−2^) at room temperature. Each pixel within the photonic barcode labels consists of closely packed WS_2_ microlasers of varying sizes, demonstrating high-density and nonuniform multiple-mode lasing signals that facilitate barcode encoding. Notably, CW operation and narrow-linewidth lasing emission could significantly simplify detection. As proof of concept, a 20-pixel label exhibits a high encoding capacity (2.35 × 10^108^). This work may promote the advancement of two-dimensional materials micro/nanolasers and offer a promising platform for information encoding and security applications.

## 1. Introduction

Micro/nano photonic barcodes have prospered in the era of big data with significant values in information security and anti-counterfeiting. Their compact size, complicated structure, and unique optical properties allow them to be used for advanced identification and make them difficult to tamper with [1,2,3,4,5]. Micro/nano photonic barcodes are commonly classified as graphical and spectral barcodes. However, the graphic encoding labels lack security due to their predictable encoding methods, making them easy to copy [6]. Furthermore, the broad and overlapping spectral signals (typically the wavelength or relative intensity of fluorescence emission) diminish identification capabilities and compromise security levels [7]. Conversely, laser-based barcodes offer clear advantages with their narrow linewidth and brightness, making them reliable for identification. Researchers are exploring micro/nanolaser-based barcodes with different structures, like Whispering Gallery Mode (WGM) and Fabry–Perot [8,9,10,11,12,13]. However, these laser systems usually need expensive pulsed lasers for detection, obstructing their practicality. So, the key is developing room-temperature continuous-wave (CW) laser-based barcodes to overcome this limitation.

Recently, two-dimensional (2D) transition metal dichalcogenides (TMDCs) have attracted significant attention because of their potential to be used as micro/nanolasers owing to their atomic-scale thickness and high emission through small material volumes [14,15,16,17,18,19]. The spatial confinement effect at the atomic level leads to an extreme localization of the carriers, whereby photons are produced with carrier concentrations quickly reaching 10^18^ cm^−3^, fulfilling the population inversion requirement for the lasing process [14]. TMDC excitonic lasers have been fabricated by integrating a TMDC monolayer with an external cavity structure, such as a WSe_2_ monolayer with a GaP photonic crystal cavity [20], a WS_2_ with a SiO_2_/TiO_2_ distributed Bragg reflector vertical cavity [15] and MoTe_2_ with a Si nanobeam cavity [21]. Nevertheless, resonant cavities require complicated microfabrication processes, which could hinder the practical applications of 2D TMDC-based lasers. In addition, all previously mentioned TMDC-based lasers utilize a single or a few layers of TMDCs as the gain medium produced by mechanical exfoliation, a pathway whose low controllability and repeatability further limit the application of micro/nanolasers.

In this paper, we report a novel high-capacity photonic barcode label by leveraging CW-pumped monolayer WS_2_ lasing. Specifically, large-area, high-quality monolayer WS_2_ films prepared via vapor deposition serve as gain mediums for external WGM resonant cavities and facilitate the construction of optically pumped microlasers. With the lensing and shielding effects of the resonant cavity, an excellent CW-pumped lasing with a full-width-half-maximum (FWHM) as low as 0.39 nm and an excitation threshold of 400 W cm^−2^ was achieved at room temperature. Furthermore, we systematically investigate the influence of resonant cavity sizes and the WS_2_ WGM lasing properties. Using a confocal microfluorescence system, we acquired high-density and nonuniform multiple-mode lasing signals suitable for barcode encoding through the lasing mapping of a photonic barcode label composed of closely packed WS_2_ microlasers with different sizes. Moreover, CW operation and narrow-linewidth lasing emission could greatly simplify detection. Remarkably, even with only 20 pixels captured, the 20-pixel label demonstrated an impressive encoding capacity of 2.35 × 10^108^. This work may promote the advancement of 2D material-based micro/nanolasers and offer a promising platform for information encoding and security applications.

## 2. Methods

### 2.1. Synthesis of Monolayer WS_2_ Films

Monolayer WS_2_ single-crystal films were grown in a tube furnace with domain-confined chemical vapor deposition (CVD) (Figure 1a). Two pieces of cleaned 300 nm SiO_2_/Si substrates formed a sandwich structure face-to-face with the mental tungsten precursor (WO_3_ powder, 3 mg) at the center and were placed together in a porcelain boat. High-purity sulfur powder (800 mg) was used as the S source. The S and the substrate zone temperatures were ~200 °C and ~770 °C, respectively. High-purity N_2_ was used as the carrier gas at a 65 mL/min flow rate, resulting in slow S precursor diffusion during the synthesis. This growth process was performed within 10 min at the grown temperature, and the furnace was left to cool naturally to room temperature.

### 2.2. Preparation of the WS_2_ Microlaser

As-grown WS_2_ films were used to fabricate WS_2_ microlasers coupled with external optical cavities. The resonant cavities with different sizes (5, 7, 10, 15 µm) were dispersed in anhydrous ethanol at a 3% volume fraction and sonicated for 10 min to ensure a uniform dispersion. Subsequently, the cavities were spin-coated onto as-grown WS_2_ films at a speed of 3000 rpm for 30 s and dried naturally.

### 2.3. Readout and Digitization of Photonic Barcode Labels

A confocal photoluminescence (PL) system was used to read photonic barcodes, and a 405 nm laser was used as the excitation source. PL mapping data were obtained for different pixels (1, 5, and 20) by moving the carrier stage. The rule for photonic barcode label identification was defined as follows: (1) setting 0.25 times the normalized PL intensity as the threshold intensity; values above and below the threshold were set to “1” and “0”, respectively; (2) the width of each bit was specified to be 2.5 nm. Finally, the emission spectra of the WS_2_ microlaser were converted to photonic barcode labels using “1” (white) and “0” (black) as symbols.

## 3. Results and Discussion

In contrast to the previously reported growth techniques for WS_2_ thin films, we successfully synthesized hundred-micron single-crystal monolayer WS_2_ in one step at 770 °C (Figure 1a), as opposed to employing a two-step process or a higher-temperature system [22,23]. The photograph and optical image of the as-grown WS_2_ sample show that our method achieved large-area WS_2_ films on silica substrates, and color contrast directly shows large, thin WS_2_ films in the central region and small, thick samples at the edges (Figure 1b,c). The as-grown monolayer WS_2_ has a regular triangular shape with a maximum edge length of 170 µm. Scanning electron microscopy images reveal that the WS_2_ films have flat, clean atomic surfaces and sharp edges, as shown in Figure 1d. We measured the thicknesses of these triangular films using the height profile obtained from atomic force microscopy (AFM) and Raman spectroscopy. The AFM image (Appendix A) indicates that the as-grown WS_2_ films are monolayers, consistent with the monolayer WS_2_ thickness reported in the existing literature [24]. The crystal structure of WS_2_ films was confirmed by transmission electron microscopy (TEM) in Figure 1e. The as-grown WS_2_ sample was transferred onto Cu grids and then characterized. The high-resolution TEM image demonstrates a hexagonal ring lattice of alternating tungsten atoms (dark dots) and sulfur atoms (gray dots). The interplanar distance of (1 0 0) was calculated to be 0.27 nm, which conforms with the 2H-WS_2_ crystal structure. Moreover, selected area electron diffraction (SAED) analysis confirmed the synthesis of monolayer single-crystal WS_2_ with semiconductor properties (Appendix A). Energy dispersive X-ray spectroscopy (EDS) and X-ray photoelectron spectroscopy (XPS) were performed to identify the composition of the as-grown WS_2_ films. The EDS elemental mappings, as depicted in Figure 1e, reveal that W and S elements are evenly distributed in the single-crystal films. Figure 1f–h present the XPS images of the as-grown WS_2_ films. The full spectra show that high-quality WS_2_ films were synthesized without impurities (Figure 1f). The fine spectra analysis of S and W led to the conclusion that the precursors were wholly transformed into WS_2_ rather than intermediate products such as WS_2x_O_2−2x_ [24]. All data unambiguously show that triangular monolayer WS_2_ films were synthesized with large-scale uniformity and high single-crystal by the CVD method. The large dimensions, perfect triangular shape, and clean surface of the monolayer WS_2_ make it an ideal candidate for gaining fundamental knowledge and developing practical applications for 2D semiconductors.

Micro-area reflectance and photoluminescence (PL) spectroscopy were employed to investigate the light emission from the as-grown WS_2_, as illustrated in Figure 2a. A prominent absorption peak at 613 nm in the reflectance curve aligns well with the PL peak of the monolayer WS_2_. Further analysis of the PL curve reveals a distinct emission peak at 613.6 nm (~2.02 eV), attributed to the neutral exciton (X^0^) according to previous studies (Figure 2b) [25,26,27,28]. In addition, a weaker luminescence peak at 624.2 nm (~1.98 eV) was identified as the charged exciton or Trion (X^T^). These favorable luminescence characteristics proved the high quality of the synthesized WS_2_ monolayer. The inset of Figure 2b presents a PL intensity mapping image of a uniform triangular WS_2_ monolayer across the sample region. The darkest area at the center may be attributed to local environmental changes, sample defects, and growth-induced strain [24]. Raman spectroscopy has been widely employed to explore molecular vibrations and rotations in 2D materials, especially for analyzing the number of layers [29,30,31], molecular doping effects [32], and internal and external strains [33]. Figure 2c,d show the Raman spectra of the as-grown WS_2_ monolayer over a frequency range from 130 cm^−1^ to 500 cm^−1^ with laser excitation at 532 nm. Typical Raman modes of WS_2_ are present, as labeled. There are first-order Raman modes, including LA (*M*), LA (*K*), E^1^_2g_ (*Γ*), E^1^_2g_ (*M*), and A_1g_ (*Γ*), and second-order modes, such as 2LA (*M*). The frequency difference between E^1^_2g_ (in-plane mode of S and W atoms) and A_1g_ (out-of-plane mode of S atoms) was 62.6 cm^−1^, confirming that the as-grown WS_2_ films were monolayer. The Raman intensity image of A_1g_ mode (inset of Figure 2c) also clearly shows the perfect triangular shape and uniform thickness of the WS_2_ films. The as-grown monolayer WS_2_, with excellent luminescent performance and uniformity, offers a fantastic material choice for micro/nano light-emitting devices.

The lasing characteristics of the WS_2_ microlaser were studied via a homebuilt micro-PL system after introducing external cavities onto the as-grown WS_2_ films to construct the microlaser, whereby a 405 nm CW laser was focused down to 5 µm and functioned as the pump source. Figure 3a shows a schematic image of the WS_2_ microlaser. The high-quality optical resonant cavity can prominently enhance the light–matter interaction of the WS_2_ gain medium, thereby improving the resonant emission of the WS_2_ microlaser. Figure 3b shows the perfect circular boundaries and the ultrasmooth surfaces of the external cavities in the WS_2_ microlaser. Dispersed resonant cavities were distributed on top of the as-grown WS_2_ films on a 300 nm SiO_2_/Si substrate, which was selected to enhance the absorption and emission efficiency because of constructive interference and a reduction in the lattice distortion [34,35]. To identify the resonance of the laser structure, Figure 3c,d illustrate the simulated electric field distribution of the WS_2_ microlaser cavity as executed by the finite element method. The analysis only considered the transverse electric (TE). According to the mode simulations, strong and weak modes are associated with fundamental modes (TE_m,1_) and higher-order modes (TE_m,2_). Figure 3c presents the electric field profile for different mode numbers. The calculated mode positions distinguished the oscillatory peaks into two groups, as shown in Figure 3d. Experimentally, a series of sharp peaks were superimposed onto the broad emission spectra, which indicates that the external resonant cavity formed typical WGM resonant feedback (Figure 3e). Moreover, the emission intensity of the WS_2_ microlaser was approximately four times higher than that of the grown substrate, as intuitively seen in the PL intensity mapping image shown in Appendix A. This phenomenon can be attributed to the lensing effect, in which excitation energy is localized at the resonant cavity and WS_2_ interface, which can significantly improve the excitation efficiency and expand spatial overlap between the excitation and gain regions [36]. Additionally, the center wavelength of the WS_2_ microlaser exhibited a red shift of ~13 meV compared with the surrounding bare monolayer WS_2_. This red shift may be attributed to screening and strain effects caused by the resonant cavity [37,38,39]. For both the WS_2_ microlaser and the bare WS_2_, the frequency separation between typical Raman feature modes of E^1^_2g_ and A_1g_ was 62.5 cm^−1^ (Appendix A), indicating that the strain effect can be ignored [40,41].

We investigated the non-linear emission characteristics of the WS_2_ microlaser by acquiring its PL spectra at various power densities ranging from 93 to 2300 W cm^−2^, as depicted in Figure 3f. At low excitation power, the PL spectrum of the microlaser closely resembled that of bare WS_2_, dominated by spontaneous emission (SE) with a broader bandwidth. As the excitation power gradually increased, a progressive narrowing of the prominent emission peaks was observed. Figure 3g presents the power dependence curve for the emission peak intensity at 625.2 nm. The curve exhibits an inflection point, signifying a transition from linear to superlinear photon density increase. This inflection corresponds to a lasing threshold of approximately 400 W cm^−2^. Below the threshold, the curve exhibits linear behavior consistent with SE. Beyond the threshold, the non-linear increase indicates amplified spontaneous emission and subsequent lasing, resulting in the characteristic S-shape. In comparison, the emission intensity of bare WS_2_ exhibits only linear SE behavior across the entire pumping range (Appendix A), further substantiating the role of the resonant cavity in enabling lasing in the microlaser. The extracted photon intensity of the lasing emission was fitted using a Lorentzian function, revealing a narrow linewidth of 0.39 nm (Figure 3h). This narrow linewidth, corresponding to a quality factor of 1560, further corroborates the occurrence of lasing action in the WS_2_ microlaser.

To systematically explore the influence of resonant cavity size on WS_2_ lasing properties, the microlasers were fabricated with purposefully varied cavity dimensions, as illustrated in Figure 4a–d. Each spectrum exhibits distinct multi-mode lasing peaks, with the number of peaks increasing as the cavity size grows. Figure 4e presents the background-subtracted emission spectra of WS_2_ microlasers with different sizes. Notably, the spacing between adjacent fundamental modes exhibits a systematic decrease, from 17.9 nm to 11.3 nm, 8.5 nm, and 5.4 nm, respectively, as the cavity size increases from 5 µm to 7 µm, 10 µm, and 15 µm. As is well known, the WGM modulation peaks follow the equation:Δλ = λ^2^/π D η_eff_(1)
where Δλ is the free spectral range (FSR) of the spacing between two adjacent same-order resonant modes. λ is the resonant wavelength. D is the diameter of the resonant cavity, and η_eff_ is the group effective index of refraction [42]. When the optical path consists of an integer multiple of the emission wavelength, the resonance emission is strengthened for the light overlap. Therefore, the resonant wavelength (λ) depends on the cavity size (D) and refractive index (η), as evident in Figure 4e. The observed variation in resonant emission wavelengths across different microlaser sizes aligns with this principle. The FSR is an α/D (where α is a constant) function of the diameters of the resonant cavities, which supports that the resonant modes can be attributed to WGM modulation [43]. Figure 4f shows the relationship between λ^2^/Δλ and D. The simulation result obtained by linear fitting reveals that the slope ηπ is 5.0, which agrees with the observed slope of 4.9. Here, η is determined to be 1.59, as π is set to 3.14.

Inspired by the unique spectral signals associated with cavity sizes, we propose a photonic barcode concept derived from the WS_2_ microlaser, schematically illustrated in Figure 5a. Owing to the distinct sizes of individual resonant cavities, the sharp lasing signals could serve as a fingerprint for each pixel, enabling their differentiation based on their characteristic emission spectra. By using a confocal microfluorescence system, we acquired high-density and nonuniform multiple-mode lasing signals, as shown in Figure 5b. Based on the spectral variations between different cavities, we encoded the lasing peaks of the WS_2_ microlaser to generate a series of corresponding photonic barcode labels.

A typical barcode label comprises a series of black bars and blanks of varying widths to represent a set of information graphically. Herein, photonic barcode labels are defined as follows: after normalizing the intensity of the WS_2_ microlaser lasing peaks, an intensity of 0.25 is specified as the intensity threshold, as shown by the horizontal dotted line in Figure 5c. All emission peaks with intensity above the threshold intensity were set to “1”, as indicated by the black bars; conversely, those below the threshold intensity were defined as “0”, as noted in the blank bars. The width of each bit was set to 2.5 nm. We intercepted each emission spectra from 610 nm to 655 nm, meaning a single pixel has 18 bits of “1”/“0”. According to the coding rule, we can deduce a specific photonic barcode label (Figure 5c) from the modulated spectra. In other words, a one-to-one correspondence between each photonic barcode label and the WS_2_ microlaser spectra exists, directly related to the dimensions of the resonant cavity. As proof of concept, we explored their application potential for anticounterfeit labels. We examined the optical performance of 5 and 20 pixels by line-by-line scanning and measuring the lasing emission of these microlasers. Figure 5d,e illustrate the acquired lasing spectra and corresponding photonic barcode labels. Surprisingly, the 20-pixel label demonstrated an impressive encoding capacity of 2.35 × 10^108^, even with only 20 pixels captured, virtually eliminating the possibility of duplication. These results suggest that photonic barcode labels utilizing CW-pumped WS_2_ microlasers could offer high sampling efficiency and enhanced security for authentication purposes, making them a promising application for anti-counterfeiting.

## 4. Conclusions

In summary, we proposed an original high-capacity photonic barcode label that utilized the CW-pumped WS_2_ lasing. The WS_2_ microlaser, based on large-area monolayer WS_2_ films, demonstrated superior lasing characteristics at room temperature, including a narrow linewidth (~0.39 nm) and a low threshold (~400 W cm^−2^) under CW pumping. Moreover, we successfully fabricated photonic barcode labels composed of a closely packed WS_2_ microlaser of varying sizes. These labels exhibit high-density and nonuniform multiple-mode lasing signals facilitating barcode encoding. As proof of concept, a 20-pixel label demonstrated a high encoding capacity (2.35 × 10^108^), indicating its potential as a platform for information encoding and security applications.

## Figures and Tables

**Figure 1 nanomaterials-14-00614-f001:**
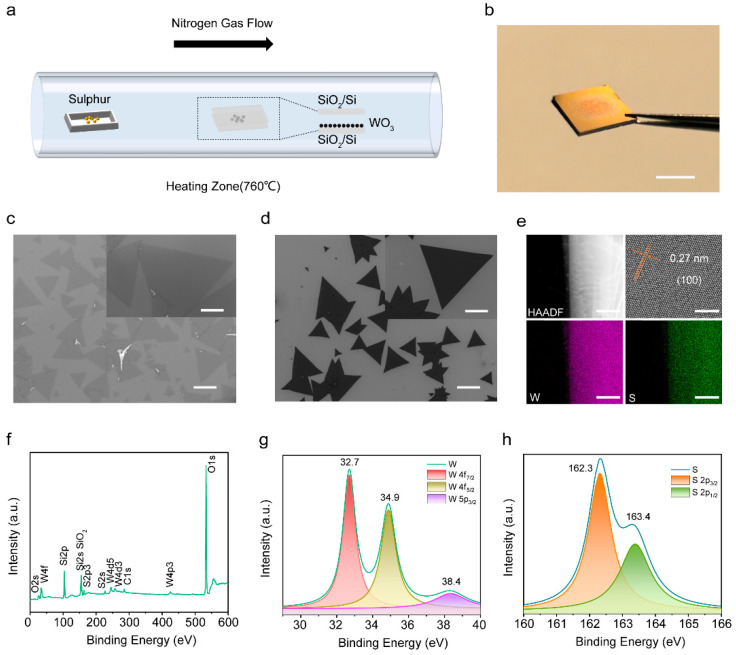
(**a**) Schematic diagram of domain-confined CVD system-grown WS_2_. (**b**) Photograph of a 300 nm SiO_2_/Si growth substrate with the as-grown WS_2_ films. Scale bar, 0.5 cm. (**c**) Optical image of the as-grown WS_2_. Scale bar, 100 µm. Inset: high magnification optical image of a triangular monolayer WS_2_. Scale bar, 40 µm. (**d**) SEM images of the as-grown WS_2_. Scale bar, 100 µm. Inset: high magnification SEM of a triangular monolayer WS_2_. Scale bar, 50 µm. (**e**) TEM images of the as-grown WS_2_ and EDS elements mapping images. The scale is 500 nm in low-resolution and 5 nm in high-resolution TEM graphs. (**f**) Full XPS spectra of the as-grown WS_2_. (**g**,**h**) XPS fine spectra of W and S elements.

**Figure 2 nanomaterials-14-00614-f002:**
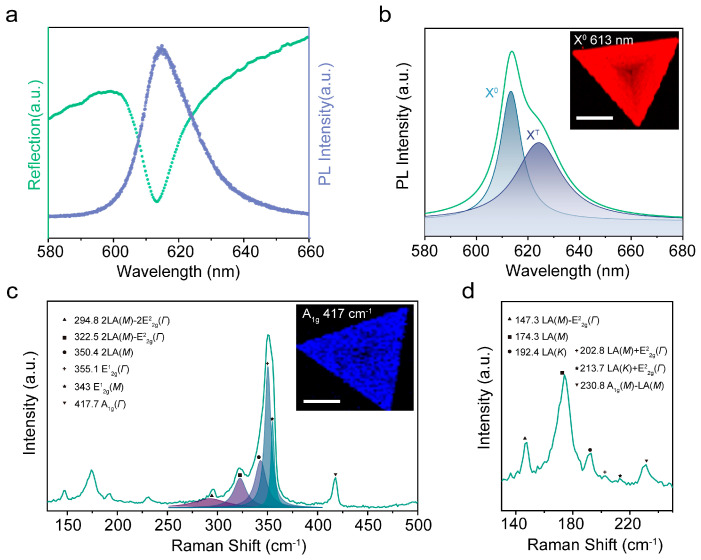
(**a**) Micro-area reflectance and PL spectra of monolayer WS_2_. (**b**) PL spectrum of monolayer WS_2_. The spectrum was divided into X^0^ and X^T^ by multi-peak fit. Inset: PL intensity mapping of a triangle monolayer WS_2_ constructed by plotting the intensity of peak X^0^. Scale bar, 20 µm. (**c**,**d**) Raman spectra of monolayer WS_2_. Inset: Raman mapping image of a triangle monolayer WS_2_ constructed by plotting the A_1g_ mode intensity. Scale bar, 20 µm.

**Figure 3 nanomaterials-14-00614-f003:**
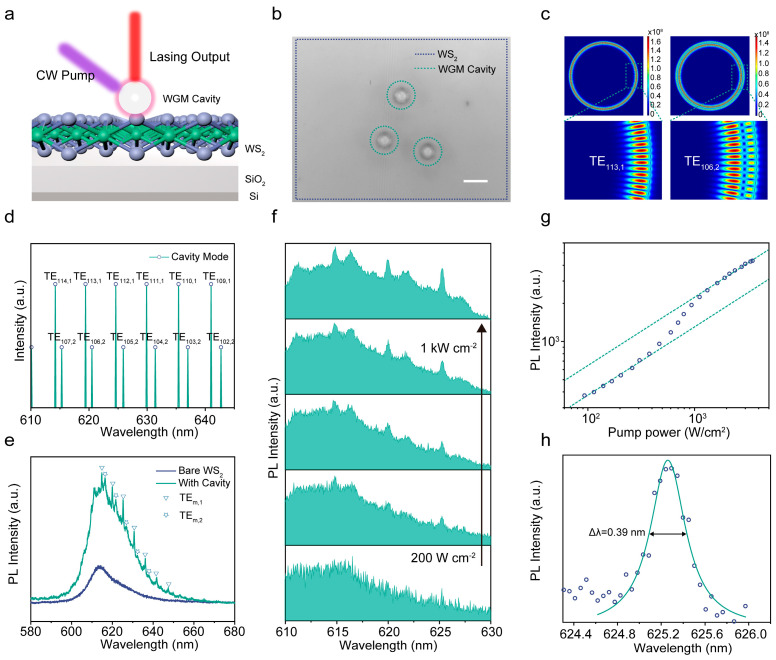
(**a**) Schematic of the WS_2_ microlaser. The red light represents laser emission from the WS_2_ microlaser. (**b**) Optical graph of the WS_2_ microlaser configuration. Scale bar, 20 µm. (**c**) Model simulation of the electric field distribution patterns of the TE_113,1_ mode at 619.76 nm (angular mode number 113) and TE_106,2_ at 620.54 nm (angular mode number 106). (**d**) Resonant cavity modes are classified into two groups via the calculated WGM positions, as indicated by the TE_m,1_ and TE_m,2_ lines. (**e**) PL spectra of the WS_2_ microlaser (dark green) and bare monolayer WS_2_ (indigo) on a SiO_2_/Si substrate at room temperature under 405 nm laser excitation (excitation power density of 500 W cm^−2^). (**f**) High-resolution emission spectra under various pump powers. The pumping power densities are as follows: 200 W cm^−2^, 400 W cm^−2^, 600 W cm^−2^, 800 W cm^−2^, and 1 kW cm^−2^, from bottom to top, respectively. The intensities of the emission peaks are normalized to clarify the character. (**g**) The plot of the integrated intensity as a function of excitation power for the TE_112,1_ mode of the WS_2_ microlaser. The indigo dashed lines are linear curves fitted to the data. (**h**) Lorentz fit for the TE_112,1_ mode emission peak of the WS_2_ microlaser above the lasing threshold.

**Figure 4 nanomaterials-14-00614-f004:**
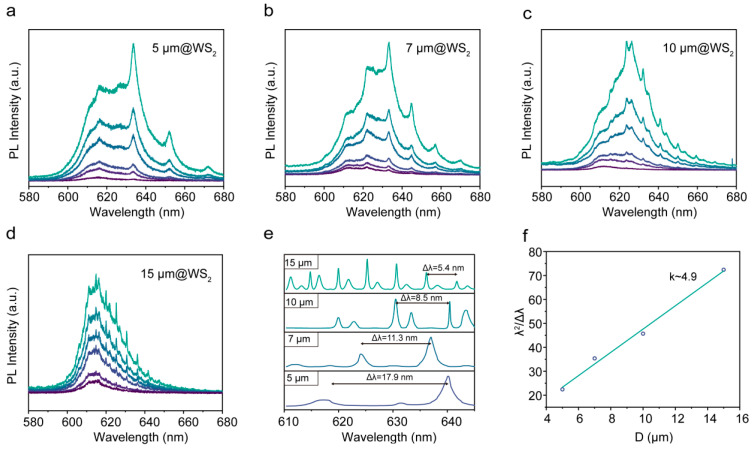
(**a**–**d**) Lasing spectra of the WS_2_ microlaser with 5 µm (**a**), 7 µm (**b**), 10 µm (**c**), and 15 µm (**d**) resonant cavities. The power density range varies from 200 W cm^−2^ to 1 kW cm^−2^, representing a progression from low to high. (**e**) The lasing spectra of the WS_2_ microlaser with four scale cavities were obtained by removing the fluorescence background for clarity. (**f**) Relationship between λ^2^/Δλ and D. The green line represents the linear fit of the data.

**Figure 5 nanomaterials-14-00614-f005:**
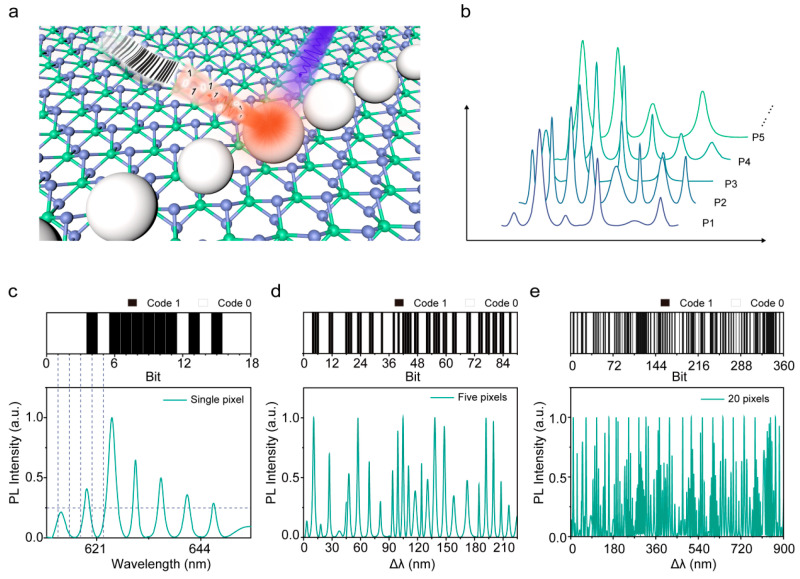
(**a**) Schematic diagram of encoding photonic barcode labels based on the WS_2_ microlaser. (**b**) The lasing spectra of different resonant cavities after removal of the fluorescence backgrounds. (**c**) Individual lasing spectrum corresponds to a photonic barcode label, with black squares specified as “1” and white squares as “0”. (**d**,**e**) Photonic barcode labels correspond to lasing spectra composed of 5 pixels (**d**) and 20 pixels (**e**).

## Data Availability

All the relevant data that support the findings of this study are available from the corresponding authors on reasonable request.

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
