# Peer review of "Continuous-Wave Pumped Monolayer WS2 Lasing for Photonic Barcoding"

_nanomaterials, 2024, doi:10.3390/nano14070614_

Round 1
Reviewer 1 Report
Comments and Suggestions for Authors
The manuscript demonstrates application of a microlasers to realize labels with high information density encoded between spectral and spatial domains. The presented microlasers were obtained by covering tungsten disulfide monolayers with whispering gallery modes resonators.
The introduction presents a clear motivation and relevant scientific background. However it would suggest expanding it by reviews about optical barcoding e.g.
https://doi.org/10.1021/acsphotonics.2c01611
https://doi.org/10.1039/D0QO00613K
The results start from the description of the CVD synthesis of WS2 monolayers. They are followed with characterisation measurements (SEM, TEM, AFM, PL, reflectance, Raman etc.) fully supporting realization of monolayer WS2 flakes.
In the following part, the authors cover the flakes with WGM resonators. With the increasing pump power the presence of the resonators results in the lasing action. The emission spectra reveals appearance of multiple narrow lines corresponding to consecutive modes of the cavities, with specific S-shape response versus the pumping power, a typical behaviour of a laser.
Later on the authors discuss the spectral spacing of the lasing lines for different sizes of the cavities.
The last part presents the premise of the paper. The authors present the idea of using spectra of the microlasers to encode information (create optical barcodes). The amount of information which can be stored can be also increased by combining data from a several microlasers.
I have some reservations to that last part. First of all the stated encoding capacity (at the order of 10108) in a significant part comes from high (18 bit) data encoded in the spectral domain. However I’m not convinced that all of such realizations are possible. I would expect (what is also presented in Fig. 3d) that the mode structure of the ring resonators is quite periodic in frequency domain, thus the spectral bits are not fully independent. Can the authors comment on that?
Also I am confused with the difference in the spectra presented in Fig. 4 and Fig. 5. The ones In Fig. 5 show significantly more lasing lines, even compared with the biggest investigated cavities in Fig. 4. What is the reason for that?
Also can the authors explain what causes the differences between the different cavities presented in Fig. 5b? Is it just a result of some variety in the cavities, or it also depends on the specific place on the flake where they are deposited (local fluctuations/strain etc.)? For practical applications in optical labelling it would be highly beneficial to be able to repeat the same label multiple times within the same or different samples. Is it possible in the proposed approach?
In summary the manuscript clearly shows realization of a room temperature microlasers and presents their interesting application for optical labelling. However before accepting the manuscript I would strongly recommend the authors to respond to my reservations about the factual encoding capacity and the repeatability of a specific label.
Apart from that I have a two remarks regarding the manuscript and several specific about data presentation.
In my opinion the “Methods” section seems misplaced. It is very succinct and hard to follow at the current place, right after the Introduction. Especially meaning of subsection 2.3 is not clear without prior reading of the whole manuscript. I would recommend moving the whole section to the end of the article.
Also surprisingly little is said about the cavities. How they were obtained? What they were made of? The information that those are WGM type resonators appears in the paper for the first time in page 5. Given that there are multiple different kinds of optical cavities I would expect finding such information earlier, if not in the abstract, at least in the introduction.
Specific comments:
· Fig. 1b: what is the scale of the image? A scale bar or some reference would be appreciated.
· Fig. 2b:The inset presents intensity of X0 peak. Does the XT follow the same behaviour? What is the size of the investigated flake?
· Fig. 3f: What are the exact pumping powers corresponding to the spectra presented in the panels? I am interested in how feasible is extracting emission intensity from a single cavity mode especially below the lasing threshold leading finally to plot in Fig. 3g. Especially as the authors themselves state “At low excitation power, the PL spectrum of the microlaser closely resembled that of bare WS2”.
· Fig. S3: what is the spatial and intensity scale? Does the four times intensity increase mentioned in the manuscript correspond to the low excitation power limit, or the powers above the lasing threshold?
· Fig. S5: What are the excitation powers used? How they compare to the range used in Fig. 3?
Author Response
Dear reviewer,
We greatly appreciate you for the professional comments and constructive suggestions on our manuscript titled “Continuous-Wave Pumped Monolayer WS2 Lasing for Photonic Barcoding” (manuscript ID: nanomaterials-2921498) together with the Supporting Information (SI) for publication in Nanomaterials. We have carefully revised the manuscript and SI according to the comments. The reviewer comments are laid out in italicized font and specific concerns have been numbered. Our response is given in normal font and modifications/additions to the manuscript and SI are given in highlight. Please see the attachment.
Best regards.
Hongxing Dong (Corresponding Author)
Key Laboratory of Materials for High-Power Laser, Shanghai Institute of Optics and Fine Mechanics, Chinese Academy of Sciences, Shanghai 201800, China
Email: hongxingd@siom.ac.cn

Reviewer 2 Report
Comments and Suggestions for Authors
In this work, Authors propose high-capacity photonic barcode labels by leveraging continuous-wave (CW) pumped monolayer tungsten disulfide (WS2) lasing. Large-area, high-quality WS2 films were grown via a vapor deposition method and coupled with external cavities to construct optically pumped microlasers, thus achieving an excellent CW-pumped lasing with a narrow linewidth and a low threshold at room temperature. According proposition of Authors, each pixel within the photonic barcode labels consists of closely packed WS2 microlasers of varying sizes, demonstrating high-density and nonuniform multiple-mode lasing signals that facilitate barcode encoding. A 20-pixel label exhibits a high encoding capacity (2.35×10^108). The work seems interesting.
Unfortunately, I do not find S1-S5 Figures anywhere about which the authors write. These Figures must be attached. Then the work should be checked again with the context of the Figures
The editor sent me the supplementary materials. I considered the manuscript again. In my opinion, the S5 drawing requires changes. Add a legend that will show the power of the laser used (color, for example yellow line - 5mW, etc.).
Figure 3f - pump power or power density should be added. After these changes, I recommend accepting the publication.Author Response
Dear reviewer,
We greatly appreciate you and the reviewers for the comments and constructive suggestions on our manuscript titled “Continuous-Wave Pumped Monolayer WS2 Lasing for Photonic Barcoding” (manuscript ID: nanomaterials-2921498) together with the Supporting Information (SI) for publication in Nanomaterials. We have carefully revised the manuscript and SI according to the comments. The reviewer comments are laid out in italicized font and specific concerns have been numbered. Our response is given in normal font and modifications/additions to the manuscript and supplementary materials are given in highlight. Please see the attachment.
Best regards.
Hongxing Dong (Corresponding Author)
Key Laboratory of Materials for High-Power Laser, Shanghai Institute of Optics and Fine Mechanics, Chinese Academy of Sciences, Shanghai 201800, China
Email: hongxingd@siom.ac.cn

Round 2
Reviewer 2 Report
Comments and Suggestions for Authors
The authors made the required changes. I believe that the article should be published